# Tolmetin Sodium Fast Dissolving Tablets for Rheumatoid Arthritis Treatment: Preparation and Optimization Using Box-Behnken Design and Response Surface Methodology

**DOI:** 10.3390/pharmaceutics14040880

**Published:** 2022-04-18

**Authors:** Mahmoud M. A. Elsayed, Moustafa O. Aboelez, Bakheet E. M. Elsadek, Hatem A. Sarhan, Khaled Ali Khaled, Amany Belal, Ahmed Khames, Yasser A. Hassan, Amany A. Abdel-Rheem, Eslam B. Elkaeed, Mohamed Raafat, Mahmoud Elkot Mostafa Elsadek

**Affiliations:** 1Department of Pharmaceutics and Clinical Pharmacy, Faculty of Pharmacy, Sohag University, Sohag 82524, Egypt; amany.abdelkader@pharm.sohag.edu.eg; 2Department of Pharmaceutical Chemistry, Faculty of Pharmacy, Sohag University, Sohag 82524, Egypt; drmoustafaaboelez@yahoo.com; 3Department of Biochemistry and Molecular Biology, Faculty of Pharmacy, Al-Azhar University, Cairo 11371, Egypt; bakheet.alkot@azhar.edu.eg; 4Department of Pharmaceutics, Faculty of Pharmacy, Minia University, Minia 61519, Egypt; hatem.sarhan@mu.edu.eg (H.A.S.); khaledak17@yahoo.com (K.A.K.); 5Medicinal Chemistry Department, Faculty of Pharmacy, Beni-Suef University, Beni-Suef 62514, Egypt; amany.mehani@pharm.bsu.edu.eg; 6Department of Pharmaceutical Chemistry, College of Pharmacy, Taif University, Taif 21944, Saudi Arabia; 7Department of Pharmaceutics and Industrial Pharmacy, College of Pharmacy, Taif University, Taif 21944, Saudi Arabia; a.khamies@tu.edu.sa; 8Department of Pharmaceutics, Faculty of Pharmacy, Delta University for Science and Technology, Gamasa 11152, Egypt; yasserhassan4543@yahoo.com; 9Department of Pharmaceutical Sciences, College of Pharmacy, AlMaarefa University, Riyadh 13713, Saudi Arabia; ikaeed@mcst.edu.sa; 10Department of Pharmacology and Toxicology, College of Pharmacy, Umm Al-Qura University, Makkah 21955, Saudi Arabia; maabdalla@uqu.edu.sa; 11Department of Pharmaceutics and Industrial Pharmacy, Faculty of Pharmacy, Merit University, Sohag 82755, Egypt; mahmoud.alkot@merit.edu.eg

**Keywords:** fast disintegrating tablets, tolmetin sodium, response surface methodology, Box-Behnken experimental design

## Abstract

Tolmetin sodium (TLM) is a non-steroidal anti-inflammatory drug (NSAIDs). TLM is used to treat inflammation, skeletal muscle injuries, and discomfort associated with bone disorders. Because of the delayed absorption from the gastro intestinal tract (GIT), the currently available TLM dosage forms have a rather protracted start to the effect, according to pharmacokinetic studies. The aim of this study was to create a combination for TLM fast dissolving tablets (TLM-FDT) that would boost the drug’s bioavailability by increasing pre-gastric absorption. The TLM-FDTs were developed using a Box-Behnken experimental design with varied doses of crospovidone (CP), croscarmellose sodium (CCS) as super-disintegrants, and camphor as a sublimating agent. In addition, the current study used response surface approach to explore the influence of various formulation and process factors on tablet qualities in order to verify an optimized TLM-FDTs formulation. The optimized TLM-FDTs formula was subsequently evaluated for its in vivo anti-inflammatory activity. TLM-FDTs have good friability, disintegration time, drug release, and wetting time, as well as fast disintegration and dissolution behavior. Significant increase in drug bioavailability and reliable anti-inflammatory efficacy were also observed, as evidenced by considerable reductions in paw thickness in rats following carrageenan-induced rat paw edema. For optimizing and analyzing the effect of super-disintegrants and sublimating agents in the TLM-FDTs formula, the three-factor, three-level full factorial design is a suitable tool. TLM-FDTs are a possible drug delivery system for enhancing TLM bioavailability and could be used to treat rheumatoid arthritis.

## 1. Introduction

In recent decades, a wide range of pharmaceutical research has been conducted in order to develop novel dosage formulations. The majority of these initiatives have been centered on medication convenience, when it comes to quality of life [1]. Orally fast dissolving tablets (FDTs) are one of the most extensively used commercial items among the dose forms intended to make drug administration easier [2]. Patients with persistent nausea, sudden episodes of allergic attacks, or coughing, such as neonatal, geriatric, bedridden, or developmentally disabled patients, may find it difficult to swallow conventional tablets or capsules, liquid orals, or syrup, resulting in ineffective therapy, with persistent nausea, sudden episodes of allergic attacks, or coughing [3]. Fast dissolving tablets could be suitable for analgesics, neuroleptics, cardiovascular agents, anti-allergies and drugs for erectile dysfunction [4]. Orally dispersible tablets (ODTs), quickly disintegrating tablets, rapid dissolve, fast melts, quick disintegrating, melt in mouth tablets, porous tablets, and freeze-dried wafers are all terms used to describe these FDTs [5]. When these formulations are placed in the mouth, saliva immediately penetrates the pores, causing the tablets to disintegrate swiftly without the patients chewing them. The time it takes for FDTs to disintegrate is typically less than one minute, with a patient’s actual disintegration time ranging from 5 to 30 s [6]. FDTs are defined as “a solid dosage form containing medicinal substances that disintegrates rapidly, usually within a matter of seconds, when placed upon the tongue” by the United States Department of Health and Human Services Food and Drug Administration Center for Drug Evaluation and Research (CDER) [7].

Non-steroidal anti-inflammatory drugs (NSAIDs) are a class of pharmaceuticals that are useful in the treatment of pain, fever, and inflammation in humans and animals [8]. TLM is generally used as a non-steroidal anti-rheumatic drug with analgesic residence in humans [9]. Broad clinical trials have set up the intensity of the medication within the treatment of adult and adolescent rheumatic joint pain and in osteoarthritis [10]. It acts by blockading the synthesis of prostaglandins (PEGs) through inhibition of cyclooxygenase enzymes (nonselective COX1 and COX2), which converts arachidonic acid to cyclic endo-peroxides, the precursors of PEGs [11]. TLM is a readily water-soluble compound that is not available in the market as an FDTs dosage form. As a result, the current study’s goal became the preparation of TLM FDTs with a rapid commencement of action. TLM-FDTs are designed using the Box-Behnken method. Box-Behnken designs are used to generate higher order response surfaces using fewer required runs than a normal factorial technique. They are still considered to be more proficient and powerful than other designs, such as the three-level full factorial design, central composite design (CCD), and Doehlert design [12]. Three components were evaluated in this model, each at three levels, and experimental trials were conducted on all 15 conceivable combinations [13]. In addition, response surface analysis is a good way to obtain a design without having to experiment for a long time. The objective of this study was to prepare TLM-FDT in order to increase TLM bioavailability, avoiding adverse effects. The effect of varying concentrations of super-disintegrants as independent variables on the percentage drug released, percentage friability, and disintegration time as dependent variables were evaluated using a Box–Behnken factorial design.

## 2. Materials and Methods

### 2.1. Materials

Biopharma Pharmaceutical Industries in Egypt generously contributed TLM. Sigma-Aldrich provided the CP and CCS carrageenan. EL-Nasr Pharmaceutical Chemicals Co., Egypt, provided anhydrous lactose, magnesium stearates, camphor, aspartame, and saccharin. E. Merck supplied the HPLC grade acetonitrile (Darmstadt, Germany). A Milli-Q Reagent Water System was used to obtain the high-quality water used to produce the solutions (Continental Water Systems, El Paso, TX, USA). All other chemicals, reagents, and solvents were purchased from regular vendors and utilized without further purification.

### 2.2. Optimization of Tolmetin Fast Dissolving Tablets Using Box-Behnken Experimental Design

#### 2.2.1. Preparation of Tolmetin Fast Dissolving Tablets

TLM-FDTs were made utilizing the direct compression method, which included super-disintegrants as well as camphor as a sublimating agent. The percentage of sublimating gent (Camphor) and the super-disintegrants added to the formulation (CP and CCS) were chosen as independent factors. The dependent (response) variables were % friability (% F), disintegration time (DT), TLM released after 10 min (Rel_10_), and wetting time (WT) [14].

All of the ingredients were sieved in a No. 60 sieve (60 mesh number and 250 µm mesh size) [9]. In a glass bottle, 50 mg of TLM was blended with the super-disintegrants (CCS and CP) for 5 min. CCS and CP were added at three distinct weight ratios to the total tablet weight: (16:150 mg) 10.6%, (20:150 mg) 13.3%, and (24:150 mg) 16.0% *w*/*w*. Geometrical dilution of the resulted TLM-super-disintegrants mixture with aspartame, camphor, saccharin, and lactose resulted in a close blend of powders. The TLM content was used to confirm mixing efficiency [10]. The resulting blend was then lubricated for 5 min with 1% *w*/*w* Mg-stearates. A single punch tablet machine (Korsch-Berlin, EK/0, Frankfurt, Germany) equipped with flat facing 8 mm punches was used to compress the mixture into tablets [10]. Camphor was sublimed in a vacuum oven at 60–65 °C for 24 h. Weighing the tablets before and after sublimation confirmed that the camphor had been removed. Table 1 shows the composition of different TLM-FDT formulations.

#### 2.2.2. Box-Behnken Design Evaluation

In Stratigraphic Plus Soft-ware, the focused response parameters were statistically examined using a one-manner ANOVA test at the level of significance (*p* < 0.05) (Stat point Tech., Inc. Warrenton, VA, USA). Furthermore, the F test and quadratic models using the following formula were used to analyze the various parameters [11,13,14,15,16]:Y = β_0_ + β_1_X_1_ + β_2_X_2_ + β_3_X_3_ + β_4_X_1_^2^ + β_5_X_1_X_2_ + β_6_X_1_X_3_ + β_7_X_2_^2^ + β_8_X_2_X_3_ + β_9_X_3_^2^(1)
where Y is the measured response for each factor-level combination, X_1_, X_2_, and X_3_ are the factors investigated, β_0_ is the intercept (constant), and 1–9 are the regression coefficients [17].

This equation allows us to investigate the effects of each element and their interactions on the response under consideration. The interaction among the primary components is represented by X_1_X_2_, X_1_X_3_, and X_2_X_3_; the quadratic terms of the unbiased variables used to approximate the curvature of the designed sample area are represented by X_12_ and X_22_ [18,19,20]. Furthermore, to fit the data into extraordinary predictor equations, a backward elimination approach was used. The three-dimensional charts were constructed using quadratic models developed using regression analysis, where the response effect (Y) was represented by a curvature surface as a characteristic of (X) [21]. To obtain the desired result, numerical optimization using the desirability technique is used to find the highest-quality settings of formula variables. Furthermore, by combining the needs of the dependent and unbiased variables, an optimized formula was created [21].

### 2.3. Characterization of TLM-FDTs

#### 2.3.1. Pre-Compression Characterization of the Powder Blend

Angle of repose, bulk density, tapped density, compressibility index (Carr’s index), and Hausner’s ratio were used to characterize the flowability qualities of powders (before compression) (Equations (2)–(6)).
(2)$$\mathrm{Angle}\;\mathrm{of}\;\mathrm{repose}=\frac{h}{r}\;$$
(3)$$\mathrm{Bulk}\;\mathrm{density}\;=\frac{M}{V_{b}}\;$$
(4)$$\mathrm{Tapped}\;\mathrm{density}\;=\frac{M}{V_{t}\;}\;$$
(5)$${\mathrm{Carr}}^{’}s\;\mathrm{Index}\;=\frac{D_{t}-{\;D}_{b}}{D_{t}}\;$$
(6)$$\mathrm{Hausner}\;\mathrm{ratio}\;=\frac{D_{t}}{D_{b}}\;$$
where h is the heap’s height, r is the base’s radius, M is the weight of the powder (gm), V_b_ is the powder’s bulk volume (mL), V_t_ is the powder’s tapped volume (mL), D_t_ is the tapped density, and D_b_ is the bulk density

#### 2.3.2. Fourier Transform Infrared Spectroscopy (FTIR)

A small amount of TLM raw material, excipients and physical mixture of TLM and all excipients were mixed with KBr pellets in a small mortar and pestle. The sample with KBr mixture paste was compressed into tablets using a hydraulic press prior to measurement of the infrared (IR) spectrum at ambient temperature. The functional groups were recorded in the frequency range of 4000–400 cm^−1^ using FTIR-8400S, Shimadzu Fourier Transform Infrared Spectrophotometer, Japan [9,22].

#### 2.3.3. Differential Scanning Calorimetry

A differential scanning calorimetry (DSC) was used to record the thermo-grams of the TLM-FDTs and their related physical mixes (DSC-60, Shimadzu, Kyoto, Japan) [22,23]. The samples were then placed in aluminum pans with punctured covers and sealed. The thermal behavior of the samples was examined at temperatures ranging from 25 to 200 °C by heating them at a rate of 10 °C/min while under nitrogen purge.

#### 2.3.4. X-ray Diffraction

X-ray diffraction (XRD) was used to study the structure of TLM and all excipients using a Lab X Shimadzu diffractometer, model XRD-6100, with nickel-filtered CuKα radiation of wavelength1.54 Å at 25 °C. The 2θ scanning range was 2° to 80° in 0.02° increments, and it was operated at 40 KV and 30 mA.

### 2.4. Post-Compression Properties of the Prepared TLM-FDTs

#### 2.4.1. Weight Variation

Twenty tablets were chosen at random for weight variation. Each formulation’s tablets were chosen and weighed using a Shimadzu digital balance. The mean and standard deviation values were calculated [24].

#### 2.4.2. Thickness

A micrometer was used to measure the thickness of ten different tablets. The mean ± SD values were calculated [25]. In packing operations and in counting tablets, filling equipment was utilized that employs the uniform thickness of the tablets as a counting mechanism. Tablet thickness becomes a crucial parameter.

#### 2.4.3. Hardness

A traditional hardness tester was used to determine the tablet’s hardness (Monsanto tablet hardness tester). Six tablets of each category were tested for hardness and results are expressed as a mean value ± standard deviation [26].

#### 2.4.4. Friability

In a plastic chambered friability apparatus (Erweka, model TA220, Heusenstamm, Germany) [9], 20 tablets of each category were accurately weighed and placed in the chamber, the plastic chamber spinning at 25 rpm and dropping the tablets at a distance of 6 inches [27]. The tablets were spun for at least 4 min in the friability tester. After the test, the tablets were dusted and reweighed, and the percentage friability was computed using the procedure below (Equation (7)) [28].
% Friability = Loss in weight/Initial weight × 100(7)

#### 2.4.5. Drug Contents

Ten tablets were weighed and powdered, and 50 mg of TLM were dissolved in 100 mL of phosphate buffer pH 7.4 with the powder. The resulting solution was filtered, diluted appropriately, then tested for drug content using a UV-Visible spectrophotometer at 324 nm (UV 324—Shimadzu, Japan) [29].

#### 2.4.6. In Vitro Disintegration Time

Disintegration test equipment [4] was used to determine the disintegration time of six tablets. Each tube of a basket with a stainless-steel bottom screen (mesh no. 10) was filled with one tablet and immersed in a water bath maintained at 37 ± 0.5 °C [4]. Using a stop watch, the time necessary for complete breakdown of the pill in each tube was calculated. Dispersible tablets must crumble within 3 min when evaluated by the disintegration test to meet pharmacopoeia standards [30].

#### 2.4.7. In Vitro Dissolution Studies

The produced tablets were tested in vitro using the USP II paddle type equipment to test the dissolution of six tablets at 37 ± 0.5 °C rotating at 50 rpm using the Electro lab dissolution tester, with phosphate buffer pH 7.4 (900 mL) as the dissolution medium. Sink conditions were maintained, and a 5 mL sample was taken at 5, 10, 15, 20 and 30 min, filtered, and spectro-photometrically measured at max 324 nm. The absorbance was measured, and % drug release was shown as cumulative percentage drug release vs. time at various time intervals [31,32].

#### 2.4.8. Wetting Time

10 mL of water containing % eosin, a water-soluble dye, was added to five spherical tissue papers in a Petri dish with a 10 cm width [9,26,33]. The colored solution was used to determine whether the tablet surface had been completely moistened. At 25 °C, the tablet was carefully placed on the tissue paper’s surface [18]. The WT [24] is the amount of time it takes for water to reach the upper surface of the tablets and completely wet them. Six replicates of these measurements were taken. A stop watch was used to record WT [18].

### 2.5. Anti-Inflammatory Activity of TLM-FDTs on Carrageenan-Induced Paw Edema in Male Wistar Rats

After receiving approval from the Ethical Committee of the Faculty of Pharmacy, Minia University, Egypt (ES15/2019), all animal experiments were carried out in accordance with the National Institutes of Health Handbook (NIH) for the Care and Use of Laboratory Animals. All in vivo experiments were carried out on healthy male Wistar rats obtained from the National Research Center (NRC) in Dokki, Giza, Egypt.

Sixteen rats were divided into four groups, each with four rats. Group I rats were used as controls. To elicit acute inflammation, rats in group II were subcutaneously injected with 0.1 mL of 1% carrageenan into the sub-plantar surface of the rat hind paw [34]. The rats in group III were given indomethacin (10 mg/kg/orally) as a conventional anti-inflammatory medicine and then subcutaneously injected with 0.1 mL of 1% carrageenan into the sub-plantar surface of the rat hind paw 30 min later [35]. TLM-FDTs (10 mg/kg/orally) were given to rats in group IV, who were then subcutaneously injected with 0.1 mL of 1% carrageenan into the sub-plantar surface of the rat hind paw 30 min later. The volume of the right hind paw was measured using a UGO-BASILE 7140 plethysmometer at intervals of 0, 5, 15, 30, 60, 120, and 180 min following carrageenan injection (Comerio, Italy). The percentage change in paw volume compared to baseline was used as the comparative criterion.

### 2.6. In Vivo Pharmacokinetic Behavior of TLM-FDTs

Six healthy male Wistar rats (weighing 150 ± 10 g) were used to assess the pharmacokinetic behavior of the TLM-FDTs. The rats were placed into two groups of three rats each and allowed to acclimate for one week under normal environmental circumstances (temperature, 22 ± 2 °C; humidity, 50 ± 5%; night/day cycle, 12 h) with free access to an ordinary rodent feed and tap water.

Animals in the first group were given TOLECTIN^®^ (TLM) tablets as a control (10 mg/kg, per os) after a 12-h fast, while animals in the second group were given TLM-FDTs (F10), with the same dose of the first group. Because of its high release percentage, F10 was chosen as an optimized formula. Blood samples were taken from each animal via the retro orbital plexus at a predefined time interval after oral dosing. The blood samples were centrifuged for 5 min at 10,000 rpm to extract plasma, which was then stored at −20 °C until analysis.

A YL9100 HPLC system (Korea) with a quaternary gradient pump, a UV detector, a G137PA automatic degasser, a standard auto-sampler, a column oven, a chromatography workstation, and a Kromasil C18 column (250 mm 4.6 mm length internal diameter, 5 m particle size) was used to determine the plasma TLM concentration in each sample. A combination of acetonitrile, methanol, and 1% acetic acid was used as the mobile phase, with a flow rate of 1 mL/min. TLM concentration in the collected plasma samples was measured at 313 nm after sample handling according to a previously established processing method [33] under these chromatographic circumstances. To precipitate proteins, 50 μL of each plasma sample were added to 50 μL of acetonitrile in an Eppendorf tube. After a 1-min vortex, each tube was centrifuged for 10 min at 10,000 rpm, and the supernatant was transferred to a glass tube containing 250 μL deionized filtered water. TLM was then extracted using 1 mL ethyl acetate. After 5 min of mechanical shaking, each tube was centrifuged for 10 min at 4000 rpm, and the supernatant was transferred to a 5 mL glass tube to be evaporated until dry in a 25 °C water bath under a nitrogen stream. A 20 μL aliquot of the acquired residue was introduced into the chromatographic apparatus after it was dissolved in 100 μL of methanol.

### 2.7. Pharmacokinetic Analysis

From each animal’s plasma concentration-time profile, pharmacokinetic parameters such as maximum plasma concentration (C_max_), time to achieve maximum plasma concentration (T_max_), absorption rate, and elimination rate constants (K_ab_, K_el_) were calculated. The trapezoidal approach was used to determine the areas under the plasma concentration-time curves from zero to the end of sampling time (AUC_0–12_). The apparent volume of distribution, half-life (T_0.5_), and total clearance (CLT) were also calculated. All of the parameters were reported as mean values with standard deviations. Using one-way analysis of variance (ANOVA) with a 95% confidence interval, the statistical significance of differences between the pharmacokinetic parameters of the tested TLM-FDTs and Tolectin^®^ tablets was evaluated.

### 2.8. Stability Study of Tolmetin Fast Dissolving Tablets

F10 was the subject of a stability investigation as it was selected as an optimized formula due to its high release percentage. The chosen formulation was kept in firmly closed bottles wrapped in aluminum foil at 30 °C/75% RH and 40 °C/75% RH [36,37] after 90 days. The drug content, weight fluctuation, % friability, hardness, wetting time, in vitro disintegration time and amount TLM released at 10 min of the stored tablets were all studied. The results were compared to those from FDTs that had been freshly manufactured.

### 2.9. Statistical Analysis

The data were statistically analyzed using the GraphPad Prism program version 6.0. (Graph Pad Software, Inc., San Diego, CA, USA). The variables were compared using one-way analysis of variance, and all values are expressed as mean SEM (ANOVA). At *p* < 0.05, differences were judged statistically significant.

## 3. Results and Discussion

### 3.1. Optimization of TLM-FDTs Using Box-Behnken Experimental Design

Box-Behnken designs have piqued the interest of academics. Box-Behnken design is slightly more efficient than the central composite design but significantly more efficient than the three-level complete factorial designs in terms of estimating quadratic model parameters and building sequential designs [38].

As a result, for the optimization study, a Box-Behnken statistical design with three factors, three levels, and 15 runs was used. A set of locations at the midpoint of each edge and the repeated center point of the multidimensional cube make up the experimental design. Table 2 lists the independent and dependent variables used in this investigation.

### 3.2. Characterization of Tolmetin Fast Dissolving Tablets Properties

#### 3.2.1. Pre-Compression Characterization of the Powder Blend

The pre-formulation study of the powder blend showed that it has low angle of repose (<30°) values, compressibility index (<35%) and Hausner’s ratio (<1.8); these indicate a good flowability of powder mixture as show in Table 3. Tablet powder possesses free flowing properties so tablets were produced of uniform weight.

#### 3.2.2. Fourier Transform Infrared Spectroscopy (FTIR)

FTIR spectra of untreated TLM for all excipients corresponded to physical mixtures (Figure 1). The spectrum of untreated TLM showed an intense characteristic peak at 1588.60 cm^−1^ that represents the ketone carbonyl (C=O) group. The presence of a peak at 1675.48 cm^−1^ indicates the carbonyl (C=O) group of carboxylic acid. The presence of a peak at 3104.03 cm^−1^ indicates sp² C–H stretching vibrations (Figure 1A) [38,39,40]. The spectra of IR spectrum of CP showed an intense characteristic peak at 1646.64 cm^−1^ that represents the carbonyl (C=O) group of β-lactam ring. The presence of peaks at 2949.11 cm^−1^, 2921.26 cm^−1^ and 2888.35 cm^-1^ indicates sp³ C-H stretching vibrations (Figure 1B). On the other hand, the spectrum of CCS showed intense characteristic peaks at 3150 cm^−1^–3450 cm^−1^ that represent the hydroxyl (O–H) groups. The presence of a peak at 1725.25 cm^−1^ indicates the carbonyl (C=O) group of carboxylic acid. The presence of peaks at 2913.22 cm^−1^ and 2874.89 cm^−1^ indicates the sp³ C–H stretching vibrations (Figure 1C) [41,42]. The spectra of Camphor showed an intense characteristic peak at 1738.77 cm^−1^ that represents the carbonyl (C=O) group. The presence of peaks at 2957.67 cm^−1^ and 2873.45 cm^−1^ indicates the sp³ C–H stretching vibrations (Figure 1D). The spectra of a physical mixture of TLM and all excipients corresponding to physical mixtures revealed the drug and carrier bands with minor shifts (Figure 1E). These results show that TLM and all of the excipients utilized have no chemical or physical interaction.

#### 3.2.3. Differential Scanning Calorimetry Study

DSC is a technique that has grown in popularity as one of the most useful thermal analysis methods for finding incompatibilities between medicinal excipients. The ordinate value at any given temperature is proportional to the differential heat flow between a sample and a reference material, according to a differential thermal analytical method. The total differential heat input is directly proportional to the integrated area under the observed curve. The endotherms represent processes in which heat is absorbed, such as melting point and phase transitions. The exotherms represent processes in which heat is evolved such as crystallization and some chemical reactions that evolve heat (Figure 2) [36,43]. The DSC thermogram of TLM alone shows a broad endothermic peak at 76.83 °C and ΔH = −230.15 J/g which corresponds to water evaporation (Figure 2A). The DSC thermogram of CP alone shows a broad endothermic peak at 90.05 °C and ΔH = −200.64 J/g which is due to the removal of some of the water trapped in the polymeric network (Figure 2B) [44]. The DSC thermogram of CCS alone shows two peaks, one sharp endothermic peak at 80.65 °C and ΔH = −93.34 J/g which corresponding to CCS melting point, the other an exothermic peak at 310.62 °C and ΔH = −792.95 J/g which is due to decomposition of CCS (Figure 2C). The DSC thermogram of camphor alone shows two endothermic peaks, one at 145.24°C and ΔH = −164.40 J/g which indicates crystalline, another endothermic peak at 175.70 °C and ΔH = −792.95 J/g which corresponds to camphor melting point (Figure 2D). The DSC thermogram of the physical mixture of TLM/all excipients shows the characteristic endothermic peaks of TLM at 77.80 °C and ΔH = −28.29 J/g, indicating that no interaction has occurred between TLM and all investigated excipients (Figure 2E).

#### 3.2.4. X-ray Diffraction Studies

The physical form of pure TLM and their related physical mixtures with other excipients was investigated using XRD (Figure 3). X-ray diffractogram of TLM alone shows a series of intense peaks at 1.22°, 15.18°, 16.43°, 19.99°, 21.26°, 25.54° and 27.84° (2θ)° which is consistent with the crystalline nature of TLM (Figure 3A) [45]. X-ray diffractogram of CP alone shows broadened peaks at 1.68°, 21.7° and 25.0° (2θ)° which suggests that there is no diffraction, i.e., no long-range three-dimensional molecular order for CP. This confirms the amorphous nature of the sample (Figure 3B) [46]. X-ray diffractogram of CCS alone which shows broadened peaks at 19.90° and 25.18° (2θ)° which is consistent with the amorphous nature of CCS (Figure 3C). X-ray diffractogram of camphor alone which shows intense peaks at 14.427°, 15.46° and 16.39° (2θ)° which is consistent with the crystalline nature of camphor (Figure 3D). X-ray diffractogram of the physical mixture of TLM/all excipients shows a series of intense peaks at 15.45°, 16.80°, 18.14°, 19.51°, 21.18°, 23.28°, 27.23° and 37.78° (2θ)°, but the intensity is lower than that in the case of TLM alone, which may be due to the dilution effect of all excipients (Figure 3E).

#### 3.2.5. Post-Compression Parameters

The prepared TLM-FDTs were determined to meet the content uniformity test criteria set out in the USP 31 specifications [18,47]. The content of TLM-FDTs in all formulations ranged from 92.20 ± 1.42% to 98.60 ± 1.66% of the theoretical label claim. The average weight variation of 20 tablets of each formula ranged from 150.5 ± 0.58 to 152.5 ± 0.63 mg, and the tablet thickness was found to be 3.0 ± 0.008 to 3.2 ± 0.12 mm. TLM-FDTs also demonstrated AV for hardness and percentage friability, with values ranging from 3.78 ± 0.32 to 4.58 ± 0.26 kg/cm^2^ and 0.229 ± 2.95% to 0.974 ± 2.02%, respectively. The WT is a critical criterion for measuring a disintegrate’s ability to swell in the presence of little water [19]. All post-compression parameters and independent variables are listed in Table 4 and Table 5. It is also worth mentioning that, as indicated in Table 5, the WT for all of the investigated formulations was less than 1 min, implying a faster disintegration time. Furthermore, with a disintegration time of 17 ± 4.12 s and a WT of 16 ± 3.65 s, formulation F9 had the quickest disintegration time. In 10 min, the tablets released 61.13 ± 2.36% (F2) and 96.1 + 3.18% (F10).

Response data for all 15 factorial design experiments (FI–F15) were collected in line with Table 5 and are shown in Figure 4, Figure 5, Figure 6 and Figure 7. There was a considerable difference between disintegration time and drug release characteristics when different combinations of factors and factor levels were used.

### 3.3. Effect of Independent Variables on the Percentage Friability

TLM-FDT formulations F1 to F15 have % friability (Y_1_) values ranging from 0.229 ± 0.295 (F14) to 0.974 ± 0.202 (F2) (Table 5). Using the ANOVA test, the effect of altering the amount of total super-disintegrants on percentage friability was examined [20,21]. The results of constructing a multiple linear regression model to characterize the relationship between % friability and the independent variables, the amount of super-disintegrants (CP and CCS), and camphor percentage were displayed in the output. The least square relapse technique was used to estimate coefficients in the approximating polynomial capacity (Equation (8)) using encoded factor level estimates [20,22]. If the effects of a factor differ significantly from zero and the *p*-value is <0.05, it is considered to have an impact on the response. A positive sign implies that the factor has a synergistic effect, whereas a negative sign indicates that the factor has an antagonistic effect on the selected response [20]. The fitted model’s equation for % friability was
Y_1_ = 1.32087 − 0.362687 X_1_ + 0.0365 X_2_ + 0.35115 X_3_ + 0.00357031 X_l_^2^ + 0.0129844 X_1_X_2_ − 0.00311719 X_2_^2^ − 0.0120625 X_2_X_3_ − 0.02215 X_3_^2^
(8)

The rise in total amounts of CP led to a decrease in the % of friability, as evidenced by the negative sign of the coefficient β_1_ (Equation (8)). The increase in total amounts of CCS and camphor result in an increase in % friability because β_2_ and β_3_ have a positive sign (Figure 4). The friability test for all tablet formulations was less than 1%, indicating that all of the tablets had acceptable mechanical resistance [23]. Tablets with low friability of less than 1% may not break during handling, packaging and/or shipping. Super-disintegrants (CP and CCS) are known to produce mechanically strong FDTs [15,16].

Figure 4 illustrates the influence of total amounts of super-disintegrates (X_1_ and X_2_) and total amounts of sublimating agent (X_3_) on the response Y_1_ in 3D response surface plots (percentage friability). Figure 4 shows that, at a medium amount of camphor and a low level of CCS, percentage friability fell from 0.974 to 0.491% when CP increased from low to high level. When CP was added to the tablet formulation, it became less friable [27]. Furthermore, at fixed CCS at medium level and camphor at low level, % friability increased from 0.463 to 0.780% when CP increased from low level (16 mg) to high level (24 mg). Moreover, at fixed X_2_ at medium level (20 mg) when X_3_ was at high level (20 mg), % friability increased from 0.783 to 0.954% when CP increased from low level (16 mg) to high level (24 mg). Increasing camphor concentration leads to an increase in the % friability, due to more porous tablets being formed, which are mechanically weak [48,49]. Finally, it can be noticed from Figure 4 that, at fixed CP at medium level and camphor at low level, % friability increased from 0.560 to 0.860% when CCS increased from low to high level. Furthermore, at fixed CP at medium level and X_3_ at high level, % friability decreased from 0.894 to 0.229% when CCS increased from low to high level. Because CCS is known to generate mechanically robust tablets, an increase in CCS concentration resulted in lower friability values. The ANOVA table partitions the variability in friability into separate pieces for each of the effects. It then tests the statistical significance of each effect by comparing the mean square against an estimate of the experimental error. In this case, one effect has P-values less than 0.05 (X_2_X_3_), indicating that they are significantly different from zero at the 95.0% confidence level.

The R-Squared statistic indicates that the model as fitted explains 75.441% of the variability in friability. The adjusted R-squared statistic, which is more suitable for comparing models with different numbers of independent variables, is 31.2349%. The standard error of the estimate shows the standard deviation of the residuals to be 0.176175. The mean absolute error (MAE) of 0.0824667 is the average value of the residuals. The Durbin-Watson (DW) statistic tests the residuals to determine if there is any significant correlation based on the order in which they occur in the data file. Since the DW value is greater than 1.4 (1.56), there is probably no serious autocorrelation in the residuals.

### 3.4. Effect of Independent Variables on the Disintegration Time

The disintegration time (Y_2_) of tablets is the most significant metric to optimize in the creation of FDT. All of the tablets in this investigation dissolved in a time range of 17 ± 4.12 s, (F9) to 26 ± 2.152 s, (F14) (Table 4). The effect of total amounts of super-disintegrates (CP and CCS) and % of sublimating agents (camphor) added to the formulation on disintegration time is best summarized by the equation below.
Y_2_ = 58.625 + 4.125 X_1_ − 10.1875 X_2_ + 2.3 X_3_ − 0.179688 X_1_^2^ + 0.171875 X_1_X_2_ − 0.0125 X_1_X_2_ 0.195312 X_2_^2^ − 0.0625 X_2_X_3_ − 0.015 X_3_^2^
(9)

The rise in total amounts of CP and camphor results in an increase in disintegration time, as evidenced by the positive sign of the coefficients β_1_ and β_3_ (Equation (9)). Because the coefficient β_2_ has a negative sign, an increase in CCS concentration causes a decrease in disintegration time. The in vitro disintegration time indicated that all of the pills have FDTs. It took less than one minute to prepare all of the pill formulations.

Figure 5 demonstrates the influence of super-disintegrates (X_1_ and X_2_) and sublimating agents (X_3_) on the reaction Y_2_ in 3D and response surface plots (disintegration time). When a larger concentration of CP was used, wicking was improved, resulting in a reduction in tablet disintegration time. The combination of wicking and swelling in the porous structure would have caused the tablets to disintegrate very quickly [27]. Furthermore, it can be observed that, at fixed X_2_ at medium level (20 mg) when X_3 is_ at low level (10 mg), Y_2_ increased from 15 s to 17 s when X_1_ increased from low level (16 mg) to high level (24 mg). Moreover, at fixed X_2_ at medium level (20 mg) when X_3_ is at high level (20 mg), Y_2_ decreased from 20 s to 19 s when X_1_ increased from low level (16 mg) to high level (24 mg). When a higher amount of camphor was employed, the tablets were predicted to have a higher porosity. Water intake increased as a result of the tablet’s porous network, and breakdown was expedited [27]. Figure 5 also shows that at fixed X_1_ at medium level (20 mg) when X_3_ is at low level (10 mg), Y_2_ increased from 19 s to 25 s when X_2_ increased from low level (16 mg) to high level (24 mg). On the other hand, at fixed X_1_ at medium level (20 mg) when X_3_ is at high level (20 mg), Y_2_ increased from 25 s to 26 s when X_2_ increased from low level (16 mg) to high level (24 mg). In high concentrations, croscarmellose sodium tends to form a viscous gel layer around the tablets, forming a barrier for further penetration of the disintegration medium into the tablet and inhibiting water penetration into the tablet core, lowering the rate of wetting and disintegration of the TLM-FDTs [29]. The effect of CP on reducing disintegration time as compared to CCS could be related to the rapid water absorbing nature of CP, which involves both capillary and swelling mechanisms that build up internal pressure, resulting in speedier disintegration [30]. Furthermore, CP has a lot of capillary activity and a lot of moisture, but it does not gel easily. With increased CP concentration in TLM-FDTs, rapid disintegration and, as a result, rapid dissolution is envisaged [31]. In this situation, four variables (X_3_, X_1_^2^, X_1_X_2_ and X_1_X_3_) have P-values of less than 0.05, suggesting that they are statistically different from zero at the 95.0 percent confidence level. According to the R-Squared statistic, the model as fitted represents almost 90.641%of the variation in disintegration time. The adjusted R-squared statistic is 73.7952 percent, which is better for comparing models with varied numbers of independent variables. The standard deviation of the residuals is 1.70294, according to the standard error of the estimate. The average value of the residuals is the mean absolute error (MAE) of 0.8. The Durbin-Watson (DW) statistic tests the residuals to determine if there is any significant correlation based on the order in which they occur in the data file. Since the DW value is greater than 1.4, there is probably not any serious autocorrelation in the residuals.

### 3.5. Effect of Independent Variables on the In Vitro Release

TLM was dissolved virtually instantly from the tablets, according to the dissolving tests. Within 10 min, about 50% of the TLM in the prepared pills was dissolved. After 10 min, the lowest and maximum amounts of TLM released (Y_3_) were found to be 61.13 ± 2.36% (F2) and 96.1 ± 3.18% (F10), respectively. The effect of total amounts of super-disintegrates (CP and CCS) and subliming agent (camphor) in TLM-FDTs formulations on in vitro released TLM at 10 min is best characterized by the following equation:Y_3_ = 33.9062 + 0.443229 X_1_ + 4.7601 X_2_ − 6.445 X_3_ + 0.163411 X_1_^2^ − 0.242031X_1_X_2_ − 0.01 X_1_ X_3_ + 0.071224 X_2_^2^ − 0.08875 X_2_X_3_ + 0.300083 X_3_^2^
(10)

The rise in total amounts of CCS and CP resulted in an increase in the TLM amount released after 10 min, as demonstrated by the positive sign of the coefficients β_1_ and β_2_ (Equation (10)). A rise in camphor concentration (from −1 to 0), on the other hand, resulted in a decrease in the amount released as indicated by the negative sign of the coefficient β_3_.

At fixed medium level of camphor and low level of CCS, TLM release increased from 61.13 to 84.8% when increasing CP concentration. Furthermore, at high level of CCS and fixed medium level of camphor, Y_3_ increased from 82.92 to 91.1% with increasing CP concentration (Figure 6). On the other hand, at fixed CCS at medium level and camphor at low level, Y_3_ increased from 76.2 to 92.5%. As the concentration of camphor rises, more pores in the tablet open up, allowing more water/saliva to enter and absorb, resulting in fast disintegration and release of the tablet’s contents. Lactose is another element in the formulations that is responsible for tablet disintegration and has a high-water solubility. The slow disintegration of particles from the tablet may be the cause of the variance in drug release from other formulations [27]. Croscarmellose sodium at high concentrations has a tendency to produce a viscous gel layer surrounding the tablets, forming a barrier to further dissolving and medium penetration into the tablet and inhibiting water penetration into the tablet core, lowering the rate of TLM-FDT release [32].

Figure 7A shows cumulative effects of drug release overall times for F1, F2, F3, F4 and F5 which are equal to 77.1 ± 1.54, 61.13 ± 2.36, 84.8 ± 1.25, 82.92 ± 2.65 and 91.1 ± 3.44, respectively, after 10 min. Figure 7B shows the cumulative effects of drug release overall times for F6, F7, F8, F9 and F10 which are equal to 76.2 ± 1.58, 92.5 ± 3.22, 75.1 ± 2.95, 80.6 ± 2.56 and 96.1 ± 3.18, respectively, after 10 min. Figure 7C shows the cumulative effects of drug release overall times for F11, F12, F13, F14 and F15 which are equal to 74.8 ± 2.19, 87.3 ± 1.25, 86.0 ± 3.22, 90.4 ± 1.25 and 76.5 ± 3.14, respectively, after 10 min.

In this case, five effects have *P*-values less than 0.05 (X_1_, X_2_, X_3_, X_1_X_2_ and X_3_^2^), indicating that they are significantly different from zero at the 95.0% confidence level. The R-Squared statistic indicates that the model as fitted explains 98.1204% of the variability in release at 10 min. The adjusted R-squared statistic, which is more suitable for comparing models with different numbers of independent variables, is 94.7371%. The standard error of the estimate shows the standard deviation of the residuals to be 2.08556. Since the DW value is greater than 1.4 (2.27), there is probably no serious autocorrelation in the residuals.

### 3.6. Effect of Independent Variables on the Wetting Time

All the tablets had WT in the range varied from 16 ± 3.65 s (F9) to 30 ± 4.12 s (F14) (Table 5), Figure 8. The effects of the total amounts of super -disintegrates (CP and CCS) added to the formulation and amounts of sublimating agents (camphor) on the WT can be best described using the following equation:Y_4_ = 92.5 + 2.27083 X_1_ - 8.44792 X_2_ − 2.725 X_3_ - 0.0364583 X_1_^2^ − 0.078125 X_1_X_2_ + 0.0375 X_1_X_3_ + 0.260417 X_2_^2^ + 0.05 X_2_X_3_ + 0.0266667 X_3_^2^
(11)

The rise in total amounts of CP resulted in an increase in the WT, as evidenced by the positive sign of the coefficient β_1_ (Equation (11)). However, because the coefficient β_2_ and β_3_ have a negative sign, an increase in CCS and camphor concentration resulted in a decrease in WT. The variability in WT is partitioned into different portions for each of the effects in an ANOVA table. The statistical significance of each effect is then determined by comparing the mean square to an estimate of the experimental error. Two effects in this case have *p*-Values < 0.05 (X_2_ and X_2_^2^), suggesting that they are statistically significant at the 95.0% confidence level. According to the R-Squared statistic, the model as fitted explains 93.2665% in WT. The corrected R-squared statistic is 81.1462%, which is better for comparing models with varying numbers of independent variables. The SD of the residuals is 1.94508 according to the standard error of the estimate. The average value of the residuals is 0.955556, which is the MAE. Since the DW value (1.228) is less than 1.4, this indicates positive autocorrelation.

WT is unaffected by raising CP from low to high levels when camphor is fixed at a medium level and CCS is low. WT fell from 29 to 24 s when CP climbed from low to high at a fixed medium level of camphor and with CCS at a high level. Furthermore, as CP climbed from low to high, WT reduced from 22 to 20 s at a set medium level of CCS and at a fixed low level of camphor. When CCS rose from low to high, WT increased from 19 to 27 s at fixed CP at medium level and a fixed high level of camphor. The hydrophilicity of excipients and the inner structure of tablets have a strong relationship with wetting time. The water infiltration rate into the powder bed is proportional to the pore radius, according to Washburn’s equation, and is impacted by the hydrophilicity of powders, which is quantified by contact angle and surface tension [34]. It is self-evident that increasing compression force and/or decreasing porosity leads to smaller pore size and, as a result, higher WT [35].

### 3.7. Anti-Inflammatory Activity of TLM-FDTs on Carrageenan-Induced Paw Edema in Male Wistar Rats

Carrageenan was injected intra-planarly into the hind paw, causing increasing paw edema; this model is used to test the anti-inflammatory activity of various medicines. The TLM-FDTs were found to have anti-inflammatory activity when compared to indomethacin as a reference medication in the current investigation. Figure 9 shows the % inhibition of the edema formation by the TLM-FDTs in comparison to indomethacin. We utilized indomethacin, a potent nonsteroidal anti-inflammatory medicine (NSAID) often used for chronic inflammatory arthritis, in the anti-inflammatory trial, and TLM tablets (TOLECTIN), which are commercially available, in the pharmacokinetic and bioavailability study. Inhibition of carrageenan-induced inflammation has been shown to be highly predictive of anti-inflammatory drug activity of TLM-FDTs over indomethacin as a result of the more rapid and efficient absorption of the TLM-FDTs that leads to rapid onset of action.

### 3.8. In Vivo Pharmacokinetics

The circulating concentrations of TLM that were observed after oral administration of TLM-FDTs and TOLECTIN^®^ (10 mg/kg) are shown in Figure 10, the normal dosing of TLM ranging from 15–30 mg/kg in normal oral dose. As the bioavailability of TLM is enhanced in our FDTs and we avoid the first pass effect which may decrease plasma concentration by about 30%, we used a loading dose of 10 mg/kg to avoid potential adverse effects caused by the elevated blood TLM concentrations and to avoid side effects. This decrease in the dose gives a competitive advantage of TLM-FDTs over conventional available dosage forms. TLM was more rapidly absorbed in the case of TLM-FDTs than the other formula due to fast release of the drug from the FDT formulation.

The specificity, linearity, accuracy, recovery, and sensitivity of the HPLC technique used were all validated. The chromatograms obtained from the blank plasma, TLM solution in blank plasma, and plasma samples taken after TLM-FDTs injection were altered using the HPLC procedure described above (Figure 10). The HPLC approach was found to be satisfactory in terms of sensitivity and specificity, with no interference with TLM determination from endogenous chemicals in plasma at the chosen chromatographic conditions, according to the results. The pharmacokinetic characteristics (Table 6 and Figure 11) revealed that peak plasma concentrations (C_max_) of TLM after administration of TLM-FDTs were substantially greater than those achieved after administration of TOLECTIN^®^ tablets (38.42 μg/mL vs. 18.40 μg/mL, respectively). TLM-FDTs had a two-fold greater area under curve (AUC) when administered in the form of fast dissolving tablets compared to TOLECTIN^®^ tablets (206.93 μg·h/mL vs. 108.58 μg·h/mL, respectively). T_max_ was also shorter in the examined FDTs than in the commercial TOLECTIN^®^ tablets (1h versus 2 h, respectively). The greater C_max_ and AUC_0–12_ values of TLM-FDTs compared to TOLECTIN^®^ tablets, as well as the shorter T_max_, indicate that TLM from the examined FDTs is more bioavailable than TLM from standard commercial tablets.

### 3.9. Stability Study

Table 7 shows the results of storing TLM-FDTs (F10 and F2) at 30 °C + 75% RH and 40 °C + 75% RH. F10 and F2 were selected as they have the highest and lowest percentage released after 10 min. During the storage period, no significant changes in the properties of kept tablets were found in the selected circumstances, according to the findings (three months). The qualities of the stored tablets were acceptable (thickness, hardness, weight variation, drug content, wetting time, in vitro disintegration time, and percentage friability). In terms of the drug release profile, the results showed that storing the developed formulations under stress for three months had a minimal effect on the rate of TLM release from TLM-FDTs, as shown graphically in Appendix A of the Appendix A.

## 4. Conclusions

A three-levels three-factors Box-Behnken Design was utilized to investigate the impact of formulation on variables such as percentage friability, disintegration time, Rel_10_, and WT, utilizing optimization techniques. The findings revealed that the levels of CP, CCS, and camphor all had an impact on the response variables. CCS, or camphor content, has a positive relationship with % friability, but CP has a negative relationship. However, there is a favorable relationship between disintegration time and CP or camphor concentration, whereas CCS has a negative impact. Additionally, there is a favorable link between the levels of TLM produced after 10 min and the concentrations of CP or CCS, although camphor has deleterious effects. Moreover, there is a negative correlation between the WT and CCS or camphor concentration, but CP has positive effects. An observed response was in close accord with the anticipated estimations of the optimized formulation and consequently shows the achievability of the improvement system in the advancement of FDTs. TLM-FDs developed in such a way as to provide an interesting field for further research given that results may be extrapolated to different medications, for which a rapid onset of action is an attractive target.

## Figures and Tables

**Figure 1 pharmaceutics-14-00880-f001:**
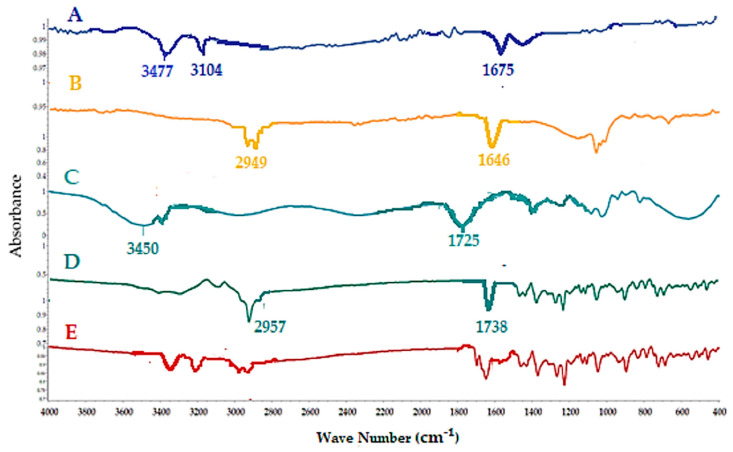
FTIR spectra of pure TLM (**A**); pure CP (**B**); pure CCS (**C**); pure camphor (**D**) and physical mixtures TLM/all excipients (**E**).

**Figure 2 pharmaceutics-14-00880-f002:**
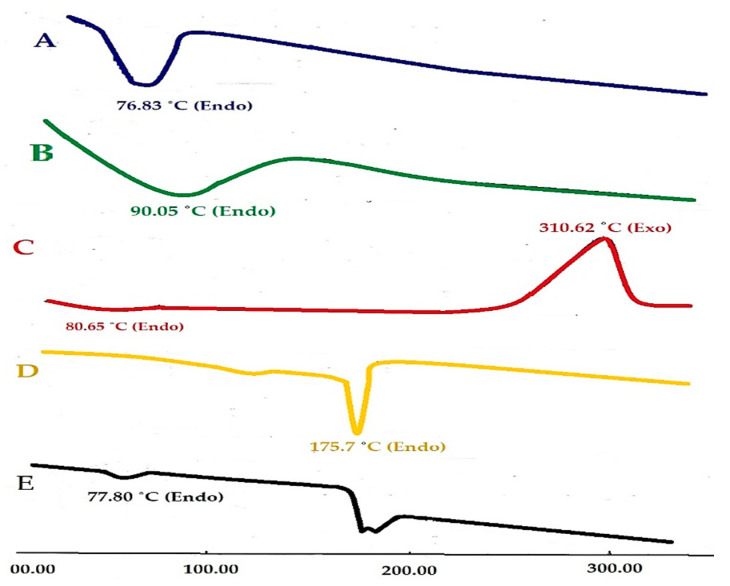
DSC thermograms of pure TLM (**A**); pure CP (**B**); pure CCS (**C**); pure camphor (**D**) and physical mixtures TLM/all excipients (**E**).

**Figure 3 pharmaceutics-14-00880-f003:**
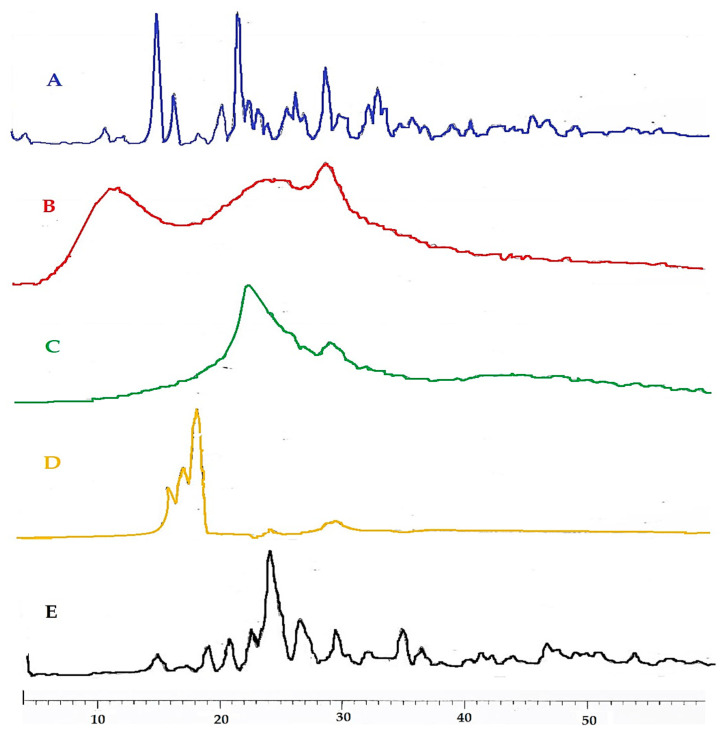
XRD profile of pure TLM (**A**); pure CP (**B**); pure CCS (**C**); pure camphor (**D**) and physical mixtures of TLM/all excipients (**E**).

**Figure 4 pharmaceutics-14-00880-f004:**
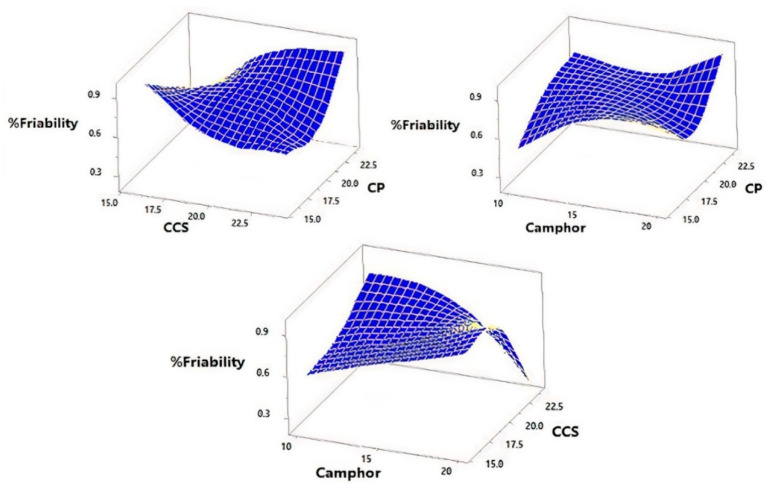
The three-dimensional contour plot for the effect of total amounts of super-disintegrates (X_1_ and X_2_) and percentage of sublimating agent (X_3_) on the percentage friability (Y_1_).

**Figure 5 pharmaceutics-14-00880-f005:**
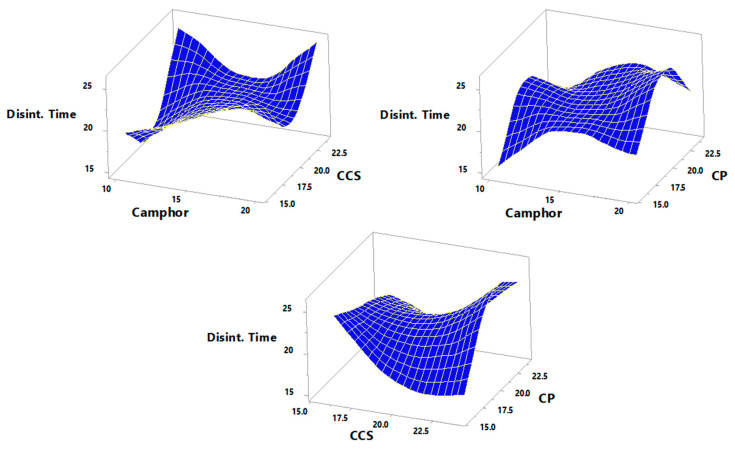
The three-dimensional contour plot for the effect of total amounts of super-disintegrates (X_1_), (X_2_) and percentage of sublimating agent (X_3_) on the disintegration time (Y_2_).

**Figure 6 pharmaceutics-14-00880-f006:**
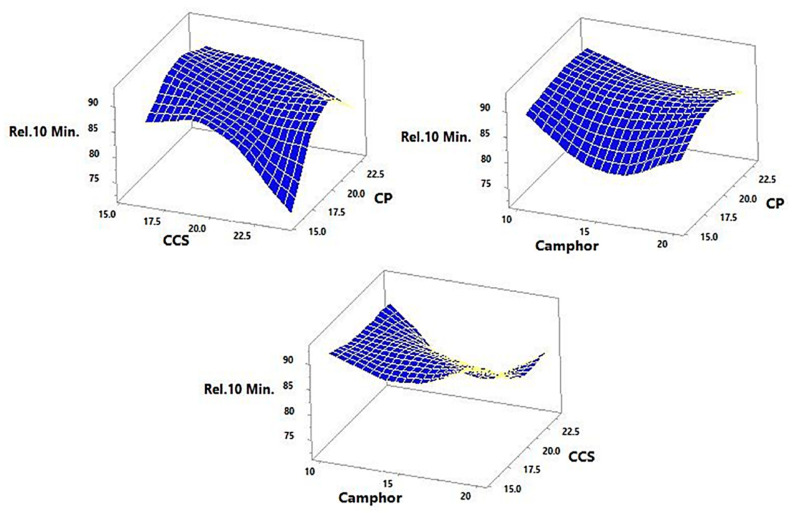
Shows 3D and response surface plots for the effect of super-disintegrates (X_1_ and X_2_) and total amount of sublimating agents (X_3_) on the response Y_3_ (% release).

**Figure 7 pharmaceutics-14-00880-f007:**
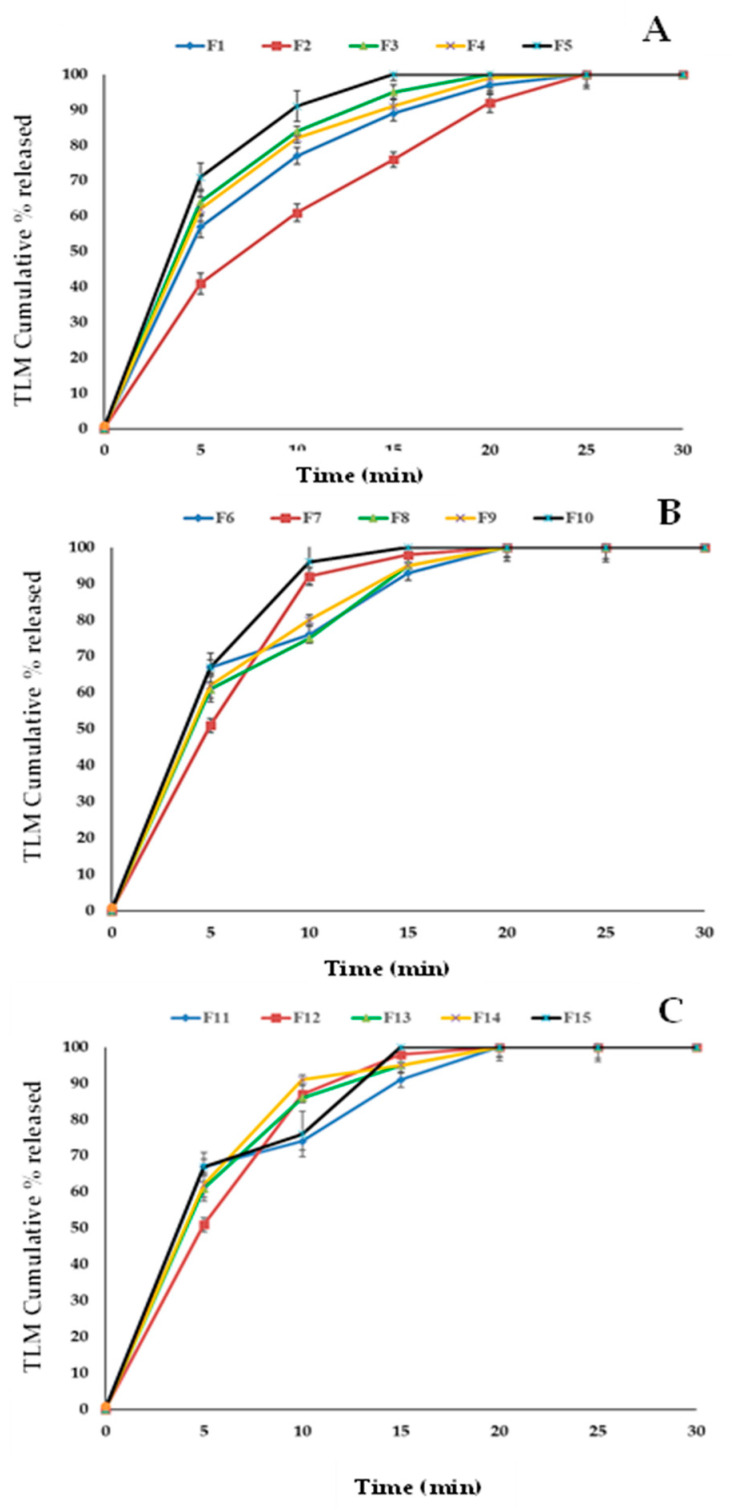
The release profiles of TLM-FDT from different formulations. Release profiles of formulae 1–5 (**A**), release profiles of formulae 6–10 (**B**), and release profiles of formulae 11–15 (**C**).

**Figure 8 pharmaceutics-14-00880-f008:**
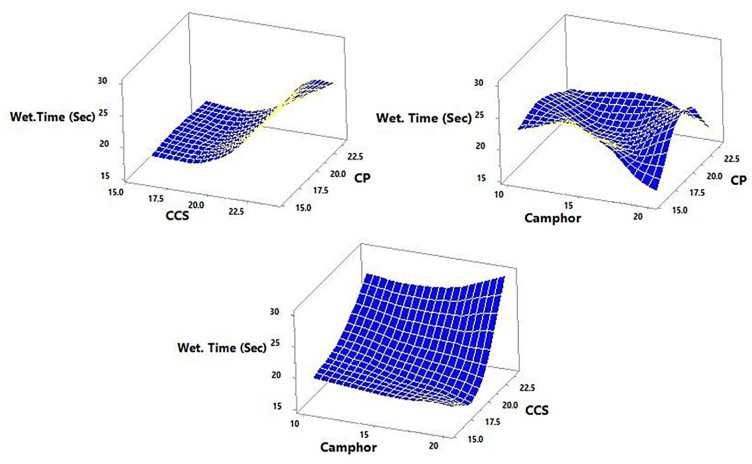
Shows the 3D and response surface plot for the effect super-disintegrates (X_1_ and X_2_) and total amount of sublimating agents (X_3_) on the response Y_4_ (WT).

**Figure 9 pharmaceutics-14-00880-f009:**
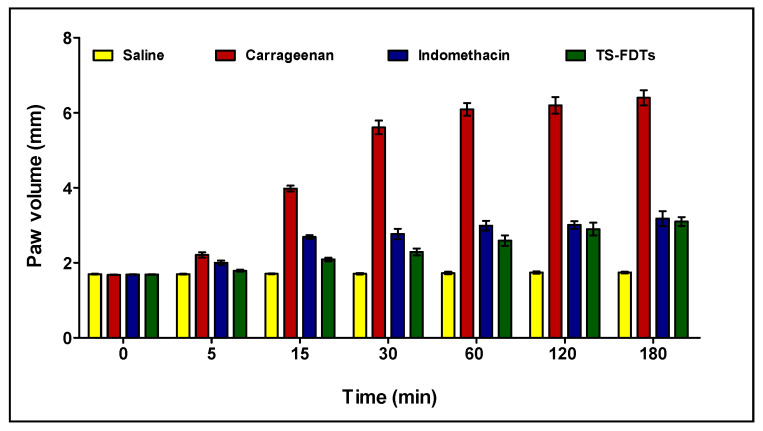
Changes in the paw volume at different time intervals in carrageenan-induced paw edema rat model. Data are presented as Mean ± SD (*n* = 6).

**Figure 10 pharmaceutics-14-00880-f010:**
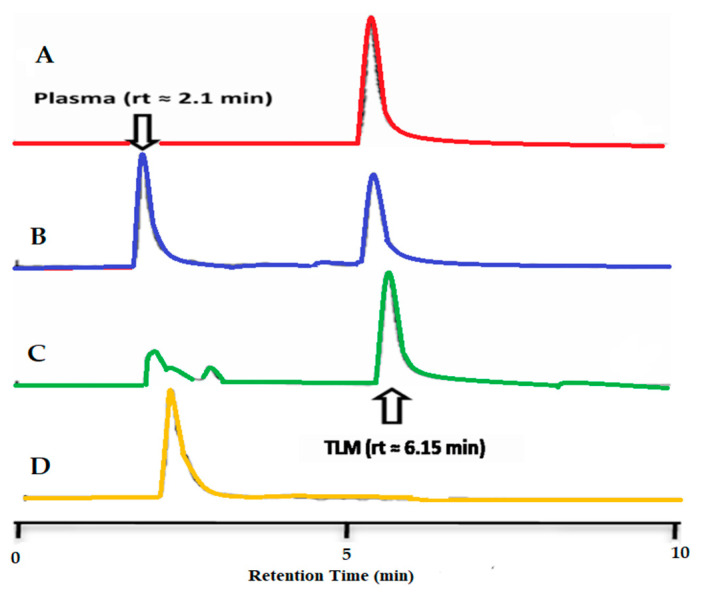
HPLC chromatograms of (**A**) TLM, (**B**) TLM in blank plasma, (**C**) plasma samples taken after injection of TLM-FDTs, and (**D**) blank plasma.

**Figure 11 pharmaceutics-14-00880-f011:**
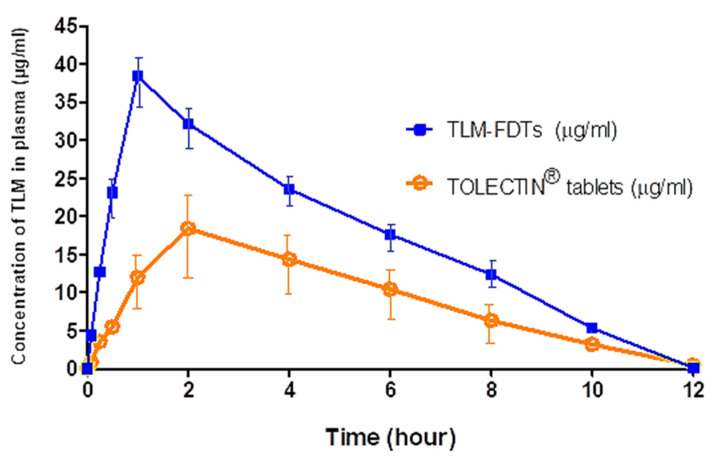
The mean concentration-time curve of TLM (μg/mL) in plasma after oral administration of TOLECTIN^®^ tablets and TLM-FDTs in rats. Data are presented as the mean ± SD (*n* = 3).

**Table 1 pharmaceutics-14-00880-t001:** Composition of different Tolmetin Fast Dissolving Tablet formulations.

Formula	TLM (mg)	CP (mg)	CCS (mg)	Camphor (mg)	Lactose (mg)	Aspartame (mg)	Saccharin (mg)	Mg-Stearate (mg)	Total (mg)
F1	50	20	20	15	37.5	4.5	1.5	1.5	150
F2	50	16	16	15	45.5	4.5	1.5	1.5	150
F3	50	24	16	15	37.5	4.5	1.5	1.5	150
F4	50	16	24	15	37.5	4.5	1.5	1.5	150
F5	50	24	24	15	29.5	4.5	1.5	1.5	150
F6	50	16	20	10	46.5	4.5	1.5	1.5	150
F7	50	24	20	10	38.5	4.5	1.5	1.5	150
F8	50	20	20	15	37.5	4.5	1.5	1.5	150
F9	50	16	20	20	36.5	4.5	1.5	1.5	150
F10	50	24	20	20	28.5	4.5	1.5	1.5	150
F11	50	20	16	10	46.5	4.5	1.5	1.5	150
F12	50	20	24	10	38.5	4.5	1.5	1.5	150
F13	50	20	16	20	36.5	4.5	1.5	1.5	150
F14	50	20	24	20	28.5	4.5	1.5	1.5	150
F15	50	20	20	15	37.5	4.5	1.5	1.5	150

**Table 2 pharmaceutics-14-00880-t002:** Box-Behnken Experimental Design.

Run Order/Batch No.	Independent Variables		
	X_1_ (mg of CP)	X_2_ (mg of CCS)	X_3_ (mg of Camphor)
F1	0 (20)	0 (20)	0 (15)
F2	−1 (16)	−1 (16)	0 (15)
F3	1 (24)	−1 (16)	0 (15)
F4	−1 (16)	1 (24)	0 (15)
F5	1 (24)	1 (24)	0 (15)
F6	−1 (16)	0 (20)	−1 (10)
F7	1 (24)	0 (20)	−1 (10)
F8	0 (20)	0 (20)	0 (15)
F9	−1 (16)	0 (20)	1 (20)
F10	1 (24)	0 (20)	1 (20)
F11	0 (20)	−1 (16)	−1 (10)
F12	0 (20)	1 (24)	−1 (10)
F13	0 (20)	−1 (16)	1 (20)
F14	0 (20)	1 (24)	1 (20)
F15	0 (20)	0 (20)	0 (15)

**Table 3 pharmaceutics-14-00880-t003:** Pre-compression characterization of the powder blend of TLM-FDTs.

Formulations	Angle of Repose (θ)	Bulk Density	Tapped Density	Carr’s Index	Hausner’s Ratio
F1	25.66 (±1.366)	0.423 (±0.009)	0.545 (±0.029)	22.38 (±1.26)	1.28 (±0.031)
F2	16.45 (±0.955)	0.420 (±0.016)	0.586 (±0.032)	28.32 (±1.42)	1.39 (±0.019)
F3	15.45 (±2.548)	0.378 (±0.015)	0.581 (±0.039)	34.93 (±1.92)	1.53 (±0.027)
F4	19.41 (±0.674)	0.381 (±0.005)	0.479 (±0.025)	20.45 (±2.54)	1.25 (±0.022)
F5	15.12 (±0.287)	0.411 (±0.022)	0.598 (±0.041)	31.27 (±0.86)	1.45 (±0.016)
F6	18.41 (±1.263)	0.409 (±0.013)	0.615 (±0.027)	33.49 (±2.11)	1.50 (±0.026)
F7	17.89 (±0.756)	0.398 (±0.019)	0.565 (±0.031)	29.55 (±3.15)	1.71 (±0.018)
F8	25.28 (±2.369)	4.20 (±0.008)	0.545 (±0.026)	22.93 (±0.56)	1.29 (±0.025)
F9	21.45 (±1.698)	0.385 (±0.011)	0.482 (±0.033)	20.12 (±3.17)	1.25 (±0.014)
F10	24.58 (±0.698)	0.412 (±0.021)	0.477 (±0.015)	13.26 (±0.96)	1.15 (±0.017)
F11	16.60 (±0.781)	0.431 (±0.019)	0.581 (±0.005)	25.81 (±1.43)	1.34 (±0.021)
F12	24.65 (±1.689)	0.422 (±0.007)	0.488 (±0.019)	13.52 (±1.77)	1.15 (±0.018)
F13	21.85 (±2.458)	0.369 (±0.009)	0.542 (±0.023)	31.91 (±2.16)	1.46 (±0.027)
F14	18.74 (±1.898)	0.358 (±0.015)	0.469 (±0.028)	23.66 (±1.06)	1.31 (±0.036)
F15	22.83 (±0.752)	4.29 (±0.007)	0.545 (±0.034)	21.28 (±1.87)	1.27 (±0.011)

*n* = 3, Loose Bulk density (g/mL) and Tapped Density (g/mL).

**Table 4 pharmaceutics-14-00880-t004:** Post- compression parameters of preliminary batches Fl-F15.

Formula No.	Thickness (mm)	Hardness (Kg/cm^2^)	Weight Variation (mg)	Drug Content (%)
F1	3.1 ± 0.08	4.02 ± 0.23	151.2 ± 0.69	95.3 ± 1.92
F2	3.2 ± 0.10	3.85 ± 0.23	152.0 ± 0.78	96.9 ± 1.50
F3	3.0 ± 0.08	4.42 ± 0.30	150.8 ± 0.82	92.2 ± 1.42
F4	3.1 ± 0.09	4.28 ± 0.22	151.1 ± 0.85	92.2 ± 1.65
F5	3.2 ± 0.11	3.88 ± 0.20	152.4 ± 0.69	96.4 ± 1.85
F6	3.0 ± 0.07	4.50 ± 0.30	150.5 ± 0.58	95.2 ± 1.74
F7	3.1 ± 0.8	4.12 ± 0.28	151.8 ± 0.58	98.0 ± 1.56
F8	3.2 ± 0.12	4.20 ± 0.29	152.3 ± 0.78	95.7 ± 1.58
F9	3.0 ± 0.9	4.15 ± 0.28	150.7 ± 0.48	98.5 ± 1.73
F10	3.1 ± 0.7	3.78 ± 0.32	151.2 ± 0.56	96.2 ± 1.65
F11	3.2 ± 07	4.34 ± 0.18	152.3 ± 0.75	96.8 ± 1.79
F12	3.0 ± 008	4.19 ± 0.24	150.8 ± 0.41	95.1 ± 1.76
F13	3.0 ± 0.13	4.12 ± 0.19	150.9 ± 0.49	96.5 ± 1.48
F14	3.2 ± 0.05	4.58 ± 0.26	152.5 ± 0.63	98.6 ± 1.66
F15	3.1 ± 006	4.18 ± 0.18	151.5 ± 0.75	95.1 ± 1.89

**Table 5 pharmaceutics-14-00880-t005:** Three factors, three levels factorial design layout showing factor combinations and response parameters of TLM-FDTs.

Formulation Number	%Friability (Y_1_)	Disintegration Time (s) (Y_2_)	Rel_10_ (%) (Y_3_)	WT (s) (Y_4_)
F1	0.815 ± 0.125	21.0 ± 2.60	77.1 ± 1.54	19 ± 4.20
F2	0.974 ± 0.202	20.0 ± 2.75	61.13 ± 2.36	18 ± 3.80
F3	0.491 ± 0.233	21.0 ± 2.76	84.8 ± 1.25	18 ± 3.90
F4	0.590 ± 0.125	25.0 ± 2.56	82.92 ± 2.65	29 ± 2.80
F5	0.938 ± 0.185	24.0 ± 3.35	91.1 ± 3.44	24 ± 3.11
F6	0.463 ± 0.224	19.0 ± 4.21	76.2 ± 1.58	22 ± 2.85
F7	0.780 ± 0.165	18.0 ± 3.54	92.5 ± 3.22	20 ± 3.44
F8	0.688 ± 0.174	20.0 ± 3.10	75.1 ± 2.95	18 ± 4.11
F9	0.783 ± 0.266	17.0 ± 4.12	80.6 ± 2.56	16 ± 3.65
F10	0.945 ± 0.141	19.0 ± 2.18	96.1 ± 3.18	17 ± 3.11
F11	0.560 ± 0.263	20.0 ± 3.65	74.8 ± 2.19	19 ± 2.54
F12	0.860 ± 0.101	25.0 ± 4.11	87.3 ± 1.25	27 ± 3.0
F13	0.894 ± 0.109	22.0 ± 3.25	86.0 ± 3.22	18 ± 3.48
F14	0.229 ± 0.295	26.0 ± 2.15	91.4 ± 1.25	30 ± 4.12
F15	0.720 ± 0.212	21.0 ± 3.62	76.5 ± 3.14	19 ± 2.59

**Table 6 pharmaceutics-14-00880-t006:** The plasma pharmacokinetic characteristics of TLM-FDTs and Tolectin^®^ tablets in rats after oral delivery.

Pharmacokinetic Parameters	TLM-FDTs	TOLECTIN^®^ Tablets
C_max_ (μg/mL)	38.42	18.40
T_max_ (h)	1	2
K_abs_ (h^−1^)	1.014	0.377
T_1/2_ (abs) (h)	0.683	1.836
AUC_0–12h_ (μg·h/mL)	206.93	108.58
AUC_0–∞_ (μg·h/mL)	206.93	108.58
Cl_T_ (mL/min)	0.791	8.414
Vd	0.100	1.246

**Table 7 pharmaceutics-14-00880-t007:** Physicochemical properties of TLM-FDTs (F10 and F2) after storage at 30 and 40 °C +RH 75% for three months compared to corresponding pre-stored tablets.

Parameters	Pre-Stored Tablets		Stored Tablets	
	Formula	Zero Time	30 °C + RH 75%	40 °C + RH 75%
Thickness (mm)	F10	3.00 ± 0.07	3.00 ± 0.21	2.95 ± 0.11
	F2	3.20 ± 0.70	3.15 ± 0.36	3.11 ± 0.25
Hardness (Kg/cm^2^)	F10	4.50 ± 0.30	4.02 ± 0.15	3.98 ± 0.21
	F2	4.34 ± 0.18	4.28 ± 0.21	4.17 ± 0.31
Weight Variation (mg)	F10	150.50 ± 0.58	150.00 ± 0.69	149.10 ± 1.20
	F2	152.30 ± 0.75	150.00 ± 0.69	149.10 ± 1.20
% Drug Content	F10	95.20 ± 1.74	94.65 ± 1.15	93.92 ± 2.03
	F2	96.80 ± 1.79	95.92 ± 1.66	95.20 ± 3.08
Wetting Time (s)	F10	22.00 ± 2.85	21.00 ± 3.54	20.00 ± 2.11
	F2	19.00 ± 2.54	18.00 ± 2.88	18.00 ± 3.25
Disintegration Time (s)	F10	19.00 ± 4.21	19.00 ± 3.25	18.00 ± 2.36
	F2	20.00 ± 3.65	19.00 ± 2.15	19.00 ± 1.58
% Friability	F10	0.46 ± 2.24	0.53 ± 1.89	0.70 ± 2.00
	F2	0.56 ± 2.63	0.60 ± 2.30	0.65 ± 1.78
% TLM released at 10 min.	F10	76.22 ± 1.58	75.03 ± 2.54	74.36 ± 4.36
	F2	74.86 ± 2.19	73.65 ± 3.69	72.69 ± 2.25

## Data Availability

Not applicable.

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
