# Peer review of "Tolmetin Sodium Fast Dissolving Tablets for Rheumatoid Arthritis Treatment: Preparation and Optimization Using Box-Behnken Design and Response Surface Methodology"

_pharmaceutics, 2022, doi:10.3390/pharmaceutics14040880_

Round 1

Reviewer 1 Report

The authors have designed an optimized formulation of fast-dissolving tolmetin tablets to treat rheumatoid arthritis and subsequently evaluated them in vivo. The result confirms a significant increase in the bioavailability of the drug and its anti-inflammatory efficacy. Here are a few small things for you to consider. 

Title is too long.

In section 3.2.4. X-ray diffraction studies, what do you mean by prepared solid dispersions?

All values of DH expressed in J/g are positives. Because the process is endothermic, tolmetin absorbs heat when it melts. Mark EXO or ENDO on the figure to more easily identify endothermic events from exothermic ones.

In the conclusions, it is pointed out that a three-level and three-factor Box-Behnken design was used to investigate the impact of the formulation with different variables, obtaining important results, however, it is not deduced which would be the best formula or if that selection is not a claims of the authors, it should be pointed out and explained in more detail, in this part, that the TLM-FD developed in this way will provide an interesting field for future research since the results can be extrapolated to different drugs.

Author Response

Many thanks for these valuable comments and suggestions and here we add our response in detail:

Comments and Suggestions for Authors

The authors have designed an optimized formulation of fast-dissolving tolmetin tablets to treat rheumatoid arthritis and subsequently evaluated them in vivo. The result confirms a significant increase in the bioavailability of the drug and its anti-inflammatory efficacy. Here are a few small things for you to consider. 

  • Title is too long.
  •  
  • Answer: Many thanks for this comment, really, we try to make it short but all our trials failed to give the message we want to deliver to the manuscript readers, so, we keep the title as it is.
  •  
  • In section 3.2.4. X-ray diffraction studies, what do you mean by prepared solid dispersions?
  •  
  • Answer: Thanks for the valuable notice and we corrected that sentence.

  • All values of DH expressed in J/g are positives. Because the process is endothermic, tolmetin absorbs heat when it melts. Mark EXO or ENDO on the figure to more easily identify endothermic events from exothermic ones.

  • Answer: Many thanks for this suggestion and we add the figure with colored one and endo and Exo marks.

  • In the conclusions, it is pointed out that a three-level and three-factor Box-Behnken design was used to investigate the impact of the formulation with different variables, obtaining important results, however, it is not deduced which would be the best formula or if that selection is not a claims of the authors, it should be pointed out and explained in more detail, in this part, that the TLM-FD developed in this way will provide an interesting field for future research since the results can be extrapolated to different drugs.

  • Answer: This is a valuable suggestion and comment, we already mentioned that F10 was selected for stability study section 2.10 and according to the comment we add this in section 2.8 In Vivo Pharmacokinetic Behavior of TLM-FDTs.

Reviewer 2 Report

Major comments

Tolmetin sodium (TLM) has been used clinically and indicated for osteoarthritis and RA, which means that the current dosage forms can bring appropriate therapeutic outcomes. Therefore, the authors must describe much more clearly the rationale for developing the orally dispersible tablet formulation of TLM.

In addition, if the oral bioavailability is increased by the developed new formulation, did the author consider the potential adverse effects caused by the elevated blood TLM concentrations?

Why did the authors choose the Box-Behnken design instead of other DoE models?

The quality of figures (1, 2, and 3) must be improved.

The weighting of the variable “disintegration time” should be increased, since the aim was to develop an ODT formulation.

Why did the authors use a 900-mL condition to evaluate the dissolution? Can this condition mimic the oral cavity with saliva?

How did the authors determine the dose (10 mg/Kg) in the 2.7. Anti-Inflammatory Activity study?

Why did the authors choose indomethacin (10 mg/kg/orally) for comparison? It seems that commercially available products of TLM should be selected.

How did the authors determine the dose (10 mg/Kg) in the pharmacokinetic study?

The extraction recovery and limit of quantification for the bioanalytical assay are vital and must be evaluated to ensure the correctness of pharmacokinetic data.

Author Response

Many thanks for these valuable comments and suggestions and here we add our response in detail:

- Tolmetin sodium (TLM) has been used clinically and indicated for osteoarthritis and RA, which means that the current dosage forms can bring appropriate therapeutic outcomes. Therefore, the authors must describe much more clearly the rationale for developing the orally dispersible tablet formulation of TLM.

  • Answer: Many thanks for this valuable comment, we mentioned that rationale in the abstract and add that in the introduction.

In addition, if the oral bioavailability is increased by the developed new formulation, did the author consider the potential adverse effects caused by the elevated blood TLM concentrations?

Answer: Many thanks for this comment which allow us to clarify that the normal dosing of TLM ranges from 15-30 mg/kg in normal oral dose. As the bioavailability of TLM is enhanced in our FDTs and we avoid the first pass effect which may decrease plasma conc. By about 30%, we used a loading dose of 10 mg/kg to avoid potential adverse effects caused by the elevated blood TLM concentrations and avoid TLM side effects. This decrease in the dose gives a competitive advantage to TLM-FDTs over conventional available dosage forms.

Why did the authors choose the Box-Behnken design instead of other DoE models?

Answer: Box-Behnken design is still considered to be more proficient and most powerful than other designs such as the three-level full factorial design, central composite design (CCD), and Doehlert design, Box-Behnken designs are used to generate higher-order response surfaces using fewer required runs than a normal factorial technique, see

The quality of figures (1, 2, and 3) must be improved.

  • Answer: We improved the quality of figures 1,2 and 3

The weighting of the variable “disintegration time” should be increased, since the aim was to develop an ODT formulation.

  • Answer: We agree with that, but the drug release may take place through diffusion or other mechanisms, so we are not focused in that point

Why did the authors use a 900-mL condition to evaluate the dissolution? Can this condition mimic the oral cavity with saliva?

  • Answer: We agree that 900 L does not mimic oral cavity conditions, our aim here was to consider the sink conditions and determine the best formula, and then process it in the pharmacokinetic and in vivo studies. In addition, the high blood supply in the oral cavity and saliva increases the drug washout, which in turn can be simulated with high receptor volume in the in vitro

How did the authors determine the dose (10 mg/Kg) in the 2.7. Anti-Inflammatory Activity study?

Answer: Many thanks for this comment which allow us to clarify that the normal dosing of TLM ranges from 15-30 mg/kg in normal oral dose. As the bioavailability of TLM is enhanced in our FDTs and we avoid the first-pass effect which may decrease plasma conc. By about 30%, we used a loading dose of 10 mg/kg to avoid potential adverse effects caused by the elevated blood TLM concentrations and avoid TLM side effects. This decrease in the dose gives a competitive advantage to TLM-FDTs over conventional available dosage forms.

Why did the authors choose indomethacin (10 mg/kg/orally) for comparison? It seems that commercially available products of TLM should be selected.

  • Answer: Many thanks for this valuable comment which we asked ourselves during running our study. Indomethacin is a potent nonsteroidal anti-inflammatory drug (NSAID) typically used for chronic inflammatory arthritis, so we used it in the anti-inflammatory study, while we used commercially available TLM tablets (TOLECTIN) for Pharmacokinetic and bioavailability study

How did the authors determine the dose (10 mg/Kg) in the pharmacokinetic study?

  • Answer: It is a comparative study, so we neglect the therapeutic dose and concentrate on the amount reached in the plasma. In addition, we try to unify the dose of 10 mg/kg throughout the study.

The extraction recovery and limit of quantification for the bioanalytical assay are vital and must be evaluated to ensure the correctness of pharmacokinetic data.

Answer: Many thanks for this valuable comment. Technically, there are differentiations between detection limits for quality and quantity from component to component, resulting from noise, response factors of instruments, and matrix interference. However, the calculation method is the cause of differentiation for each component of the different methods. The results show that no matter what component, the relationship between these levels in different methods is approximately

Reviewer 3 Report

This is a well-written paper, on a relevant subject, and deserves publication. I really like the detailed study and indication of error margins, which is so often neglected nowadays.

On page 11 I think it reads strange ‘The average per-centage deviation of 20 tablets of each formula was less than 1507.5 mg,’  A percentage is not in mg, and 1507 mg seems a very high number in view of the mass of individual tablets. Please check carefully.

Also page 11 ‘AV for hardness and percent friability, with values ranging from 3.85±0.23 to 4.58±0.26 kg ‘  Please add Figure 4 after hardness and Figure 5 after friability. Also the unit is not kg but kg/cm2.

Further on page 11 ‘as indicated in Table 4, the WT’  I think this should be Table 5.

Figure 2, second plot, the name along one of the axis cannot be read (because of the overlapping 3rd plot), please correct.

Below Figure 4, in the text you describe what is to be seen in Figure 4, you wrote ‘Figure 4 shows that at a medium amount of camphor and a low level of CCS, percent friability fell from 0.974% to 0.491% when CP increased from low to high level. When CP was added to the tablet formulation, it became less friable [27].’ I do not doubt that this is correct, but it is hard to see because one sees two parameters (to describe friability) but the value of the third one is not mentioned. So I find it difficult to see what you state with respect to the three parameters CCS, Camphor and CS.

There is Supplementary Material, where is it referenced in the manuscript?

Author Response

Many thanks for these valuable comments and suggestions and here we add our response in detail:

  • On page 11 I think it reads strange ‘The average percentage deviation of 20 tablets of each formula was less than 1507.5 mg,’  A percentage is not in mg, and 1507 mg seems a very high number in view of the mass of individual tablets. Please check carefully.
  •  
  • Answer: The percentage reviewed and corrected

Also page 11 ‘AV for hardness and percent friability, with values ranging from 3.85±0.23 to 4.58±0.26 kg ‘  Please add Figure 4 after hardness and Figure 5 after friability. Also, the unit is not kg but kg/cm2.

  • The units and figures were added and corrected

Further on page 11 ‘as indicated in Table 4, the WT’  I think this should be Table 5.

  • Answer: Corrected

In figure 2, the second plot, the name along one of the axis cannot be read (because of the overlapping 3rd plot), please correct this.

  • Answer: We corrected that and insert colored figures with clear legends and plot area

Below Figure 4, in the text you describe what is to be seen in Figure 4, you wrote ‘Figure 4 shows that at a medium amount of camphor and a low level of CCS, percent friability fell from 0.974% to 0.491% when CP increased from low to a high level. When CP was added to the tablet formulation, it became less friable [27].’ I do not doubt that this is correct, but it is hard to see because one sees two parameters (to describe friability) but the value of the third one is not mentioned. So I find it difficult to see what you state with respect to the three parameters CCS, Camphor, and CS.

Answer: We agree that it is difficult to see that, so we used to illustrate and, in our figures, there are three dimensions X, Y, and Z. any point in the plot area has these three values through the 3 axes and this is the value of the response surface figures as it was used for formulations optimization.

There is Supplementary Material, where is it referenced in the manuscript?

  • Answer: We already referenced them in section 3.9 Stability study ( 1S and 2S)

Round 2

Reviewer 2 Report

The authors have replied to the questions accordingly. However, I would suggest that the authors could select some content of answers in "Author's Notes", and add it to the Discussion section of the manuscript, which could help readers to understand the reasons for experimental parameter settings, which can also much further elevate the value and readability of this paper.   

Author Response

Many thanks for these valuable comments and suggestions and here we follow your suggestions, here is our response:

  • The authors have replied to the questions accordingly. However, I would suggest that the authors could select some content of answers in "Author's Notes", and add it to the Discussion section of the manuscript, which could help readers to understand the reasons for experimental parameter settings, which can also much further elevate the value and readability of this paper.
  •  
  • Many thanks for this valuable suggestion, we select some content of answers in "Author's Notes", and add them to the introduction and discussion sections of the manuscript. (Introduction. Sections 3.7 and 3.8)

This manuscript is a resubmission of an earlier submission. The following is a list of the peer review reports and author responses from that submission.